# An Interpretable Radiomics Model Based on Two-Dimensional Shear Wave Elastography for Predicting Symptomatic Post-Hepatectomy Liver Failure in Patients with Hepatocellular Carcinoma

**DOI:** 10.3390/cancers15215303

**Published:** 2023-11-06

**Authors:** Xian Zhong, Zohaib Salahuddin, Yi Chen, Henry C. Woodruff, Haiyi Long, Jianyun Peng, Xiaoyan Xie, Manxia Lin, Philippe Lambin

**Affiliations:** 1Department of Medical Ultrasonics, Institute of Diagnostic and Interventional Ultrasound, The First Affiliated Hospital of Sun Yat-sen University, Guangzhou 510080, China; zhongx63@mail.sysu.edu.cn (X.Z.); longhy9@mail.sysu.edu.cn (H.L.); pengjy53@mail2.sysu.edu.cn (J.P.); xiexyan@mail.sysu.edu.cn (X.X.); 2The D-Lab, Department of Precision Medicine, GROW—School for Oncology and Reproduction, Maastricht University, 6220 MD Maastricht, The Netherlands; z.salahuddin@maastrichtuniversity.nl (Z.S.); yi.chen@maastrichtuniversity.nl (Y.C.); h.woodruff@maastrichtuniversity.nl (H.C.W.); philippe.lambin@maastrichtuniversity.nl (P.L.); 3Key Laboratory of Intelligent Medical Image Analysis and Precise Diagnosis, College of Computer Science and Technology, Guizhou University, Guiyang 550025, China; 4Department of Radiology and Nuclear Medicine, GROW—School for Oncology and Reproduction, Maastricht University Medical Center+, 6229 HX Maastricht, The Netherlands

**Keywords:** hepatocellular carcinoma, post-hepatectomy liver failure, two-dimensional shear wave elastography, radiomics, interpretability

## Abstract

**Simple Summary:**

Two-dimensional shear wave elastography (2D-SWE) has demonstrated predictive value for symptomatic post-hepatectomy liver failure (PHLF) in hepatocellular carcinoma (HCC). Our aim was to develop and validate an interpretable radiomics model based on 2D-SWE for predicting symptomatic PHLF in patients undergoing liver resection for HCC. We proposed an interpretable clinical–radiomics model based on both multi-patch radiomics and clinical features, which showed an AUC of 0.822 in the test cohort, higher than the clinical model (AUC: 0.684, *p* = 0.007), radiomics model (AUC: 0.784, *p* = 0.415), end-stage liver disease (MELD) score (AUC: 0.529, *p* < 0.001), and albumin–bilirubin (ALBI) score (AUC: 0.644, *p* = 0.016). The SHAP analysis showed that first-order radiomics features were the most important features for PHLF prediction. The clinical–radiomics model is useful for predicting symptomatic PHLF in HCC with high model interpretability, which may serve as a useful tool for therapeutic decision making to improve perioperative management.

**Abstract:**

Objective: The aim of this study was to develop and validate an interpretable radiomics model based on two-dimensional shear wave elastography (2D-SWE) for symptomatic post-hepatectomy liver failure (PHLF) prediction in patients undergoing liver resection for hepatocellular carcinoma (HCC). Methods: A total of 345 consecutive patients were enrolled. A five-fold cross-validation was performed during training, and the models were evaluated in the independent test cohort. A multi-patch radiomics model was established based on the 2D-SWE images for predicting symptomatic PHLF. Clinical features were incorporated into the models to train the clinical–radiomics model. The radiomics model and the clinical–radiomics model were compared with the clinical model comprising clinical variables and other clinical predictive indices, including the model for end-stage liver disease (MELD) score and albumin–bilirubin (ALBI) score. Shapley Additive exPlanations (SHAP) was used for post hoc interpretability of the radiomics model. Results: The clinical–radiomics model achieved an AUC of 0.867 (95% CI 0.787–0.947) in the five-fold cross-validation, and this score was higher than that of the clinical model (AUC: 0.809; 95% CI: 0.715–0.902) and the radiomics model (AUC: 0.746; 95% CI: 0.681–0.811). The clinical–radiomics model showed an AUC of 0.822 in the test cohort, higher than that of the clinical model (AUC: 0.684, *p* = 0.007), radiomics model (AUC: 0.784, *p* = 0.415), MELD score (AUC: 0.529, *p* < 0.001), and ALBI score (AUC: 0.644, *p* = 0.016). The SHAP analysis showed that the first-order radiomics features, including first-order maximum 64 × 64, first-order 90th percentile 64 × 64, and first-order 10th percentile 32 × 32, were the most important features for PHLF prediction. Conclusion: An interpretable clinical–radiomics model based on 2D-SWE and clinical variables can help in predicting symptomatic PHLF in HCC.

## 1. Introduction

Hepatocellular carcinoma (HCC) ranks as the fifth most common malignancy and the third leading cause of cancer-related death globally [1]. Liver resection serves as the primary curative approach for eligible HCC patients [2]. Despite advances in surgical techniques and perioperative care, post-hepatectomy liver failure (PHLF) remains the predominant factor behind postoperative morbidity and mortality, with an overall incidence of up to 32% and corresponding mortality of up to 5.0% [3]. Moreover, PHLF occurs in the first few days after liver resection, which may necessitate some additional interventions [4]. Thus, preoperative prediction of PHLF is of great importance to improve perioperative management, optimize treatment options, and avoid life-threatening events during liver resections.

PHLF primarily affects patients with liver cirrhosis who have a limited capacity for liver regeneration and diminished functional reserve of the remaining liver following resection [5]. Therefore, it is crucial to accurately assess preoperative liver functional reserve for the prediction of PHLF. Several liver function indicators, including the Child–Pugh score, model for end-stage liver disease (MELD) score, albumin–bilirubin (ALBI) grade, and indocyanine green clearance (ICG) test, have been proposed for PHLF prediction, albeit with limited accuracy, with the areas under the receiver (AUCs) operating characteristic curve ranging from 0.61 to 0.76 [6,7,8,9]. Two-dimensional shear wave elastography (2D-SWE) is an innovative liver stiffness measurement (LSM) technology that combines B-mode ultrasound imaging with real-time color-coded tissue stiffness mapping [10], and 2D-SWE has demonstrated excellent performance in assessing the degree of liver fibrosis [11]. Previous studies have also highlighted the potential value of LSM using 2D-SWE in PHLF prediction [12,13]. However, routine analyses of 2D-SWE fail to fully utilize all information available in the images and also suffer from inter-observer variance in choosing the optimal quantification region [14]. A computer-aided quantitative analysis of 2D-SWE images may help overcome these limitations [15].

Radiomics is the high-throughput extraction of quantitative features from medical imaging, converting these into minable data, which can then be analyzed for use in decision support systems [16,17]. Radiomics has shown great potential for the quantitative analysis of SWE images [15,18,19]. Several studies have shown that radiomics models of 2D-SWE images showed a good performance in the classification of liver fibrosis [15,20]. However, no previous study has evaluated the utility of radiomics for the analysis of 2D-SWE images for predicting symptomatic PHLF in patients with HCC.

Despite significant progress in radiomics, the clinical translation of artificial intelligence (AI) tools has so far been limited, partially due to a lack of interpretability of models, the so-called “black box” problem [21]. Model interpretability is important for clinicians to understand the models. Post hoc interpretability methods such as Shapley Additive exPlanations (SHAP) can be used to gain insight into the decision-making process of complex classifiers in radiomics [22]. The SHAP interpretability method calculates the significance of each radiomics feature, which helps the doctors understand the model.

Thus, this study aimed to evaluate the feasibility of radiomics model based on 2D-SWE for predicting symptomatic PHLF in patients undergoing liver resection for HCC. Furthermore, we studied the utility of SHAP for the interpretability of the radiomics model.

## 2. Materials and Methods

### 2.1. Patients

The protocol of this prospective study was approved by the Institutional Review Board of the First Affiliated Hospital of Sun Yat-sen University in China. Written informed consent was obtained from all patients before their enrollment. Patients who were candidates for curative liver resection for HCC between August 2018 and October 2022 were enrolled in this study. The diagnosis of HCC was determined according to the American Association for the Study of Liver Diseases (AASLD) Clinical Practice Guidelines for HCC (Edition 2018) [23], and the staging of HCC was determined in accordance with Barcelona Clinic Liver Cancer (BCLC) staging (Edition 2018) [24]. The inclusion criteria were as follows: (1) patients with resectable and treatment-naive HCC and (2) patients with a performance status Eastern Cooperative Oncology Group (PS) score of 0–1. The exclusion criteria were as follows: (1) patients who did not undergo liver resection; (2) patients with a pathological diagnosis of non-HCC; (3) failure in liver stiffness measurement defined as the elastography color map was less than 75% filled or an interquartile range (IQR)/median > 30%; (4) patients with evidence of immune-active chronic hepatitis characterized by an elevation of alanine aminotransferase (ALT) levels ≥ 2 × upper limit of normal (ULN); (5) patients experiencing obstructive jaundice or the presence of intrahepatic bile ducts dilation with a diameter of >3 mm; and (6) patients with hypoalbuminemia, hyperbilirubinemia, or coagulopathy not related to the liver. Figure 1 shows the patient recruitment process. Patients enrolled from August 2018 to February 2021 were the training cohort, while patients enrolled from March 2021 to September 2022 were the test cohort.

### 2.2. Two-Dimensional SWE Data Acquisition

Patients underwent 2D-SWE examination within one week before surgery. A single radiologist (M.L) with more than 10 years of experience in liver ultrasound examination and more than 3 years of experience in liver 2D-SWE examination performed the examination. The radiologist was blinded to the clinical status of each patient.

The 2D-SWE examination was performed using the SuperSonic Imagine Aixplorer™ ultrasound system with Real-time ShearWave™ Elastography (SWE™) technique using a convex broadband probe (SC6–1, 1–6 MHz). Firstly, a B-mode ultrasound scan was performed to identify a suitable liver area for 2D-SWE measurement, which was well visualized, free of large vessels, and located at least 5 cm away from any lesion. Areas in the right lobe of the liver were preferred if available. When an appropriate area was located, the B-mode ultrasound mode was switched to elasticity imaging mode. The scale was set as 40 kPa, and the depth was set at 4–6 cm. The 2D-SWE box was set to 4 × 3 cm in size and was positioned 1.5–2 cm beneath the liver capsule. Patients were asked to hold their breath for 4–5 s to obtain a series of 3–10 consecutive 2D-SWE images. All images were stored in the Digital Imaging and Communications in Medicine (DICOM) format. Color filling in the 2D-SWE box that reached more than 75% was considered successful. A circular region of interest (ROI, termed Q-box) of 2 cm in diameter was placed on the most homogeneous area assessed visually to derive the mean value of elasticity. Independent mean values were obtained from each elastography image for each patient, and the median and interquartile range (IQR) values were calculated. The 2D-SWE image quality criteria were set at IQR/median < 30% [10].

### 2.3. Clinical Data Collection

Preoperative patient characteristics; laboratory data; and radiological data, including upper abdominal computed tomography (CT) and magnetic resonance imaging (MRI), were collected within one week before surgery. Clinically significant portal hypertension (CSPH) was defined as the presence of esophageal varices (by CT/MR) and/or platelet count <100 × 10^9^/L in association with splenomegaly [25]. Splenomegaly was defined as the longest diameter of the spleen greater than 12 cm measured on coronal and axial CT/MRI images in the portal venous phase [26]. The Child–Pugh score, ALBI score, and MELD score were calculated according to formulas presented in Appendix A. Total liver volume (TLV), resected liver volume (RLV), and future liver remnant volume (LRV) were assessed based on 3-dimensional reconstruction and simulation of surgical resection plan on preoperative CT or MRI imaging. LRV ratio was defined as liver remnant volume/total liver volume to represent the percentage of the remnant liver after resection.

### 2.4. Diagnosis and Staging of Symptomatic PHLF

The definition of PHLF followed the guideline proposed by the International Study Group of Liver Surgery (ISGLS), which defined it as an increased international normalized ratio (INR) and hyperbilirubinemia on or after postoperative day 5 [3]. The severity of liver failure was categorized based on its impact on clinical treatment. Patients with PHLF grade A required no change in clinical treatment. For patients with PHLF grade B, there was a deviation from the standard treatment, but invasive therapy was not necessary. Patients with PHLF grade C required invasive therapeutic interventions. The symptomatic PHLF group was defined as those with PHLF grade B or higher, while the non-symptomatic PHLF group included individuals with PHLF grade A or those without the presence of PHLF [27].

### 2.5. Construction of Radiomics Models

The workflow of the construction of radiomics models is presented in Figure 2.


(1)Image preprocessing:


A four-step process was used for preprocessing the elasticity data. First, the 2D-SWE box was automatically extracted from the DICOM images, which is a combination of elastographic images and B-mode images. The original color elasticity image was obtained by subtracting 50% of the corresponding B-mode image from the combined image, and the color elasticity image was resized to 128 × 128 pixels. Second, the circular measurement marks in the 2D-SWE images indicating the location of the Q-box were detected and replaced with the mean value of the surrounding 4 × 4 pixels. Third, the hue-match method was used for converting RGB color elasticity images to gray images [28]. The raw elasticity data were encoded into color images according to the color bar displayed on the DICOM image, which had 220 pseudo-color levels from blue to red, representing elasticity modulus values from 0 to the maximum measurement (Figure 3a). The color bar was linearly subdivided into 2200 color levels, and the RGB value of the k-th level was denoted by (R_k_, G_k_, and B_k_). The hue value of (R_k_, G_k_, and B_k_) was computed as H_k_ = arctan(2R_k_-G_k_-B_k_,√3(G_k_-B_k_)). For a particular pixel of the color elasticity image, its hue value was computed as H_e_ = arctan(2R-G-B,√3(G-B)), where (R, G, and B) were the RGB value of the pixel. We found the index 1 ≤ k ≤ 2200 that minimized the difference |H_k_-H_e_|, and the k*maximum measurement/2200 was calculated as the reconstructed elasticity data of this pixel. After pixel-by-pixel reconstruction, the color elasticity image in the RGB space (Figure 3b) was transformed into a gray image (Figure 3c) whose values varied from 0 to maximum measurement. The hue-match method was compared with other reported methods of RGB-to-gray SWE image conversion, including distance match [18], RGB three-channel methods [19], and direct conversion from RGB to gray via a formula [19]. The hue-match method was chosen because of its superior performance when compared with the other methods in terms of the AUC. Fourth, an automated ROI selection of different size patches (32 × 32, 64 × 64, and 96 × 96) was performed by scanning 32 × 32, 64 × 64, and 96 × 96 pixel ROIs over the 2D-SWE image at 1-pixel spacing to produce numerous candidate ROIs. For each scale, the ROI with the smallest standard deviation (SD) of the pixel values within all candidate ROIs was selected for further analysis [29].


(2)The radiomics model based on 2D-SWE images:


Radiomics features were automatically extracted from different patches of ROI (32 × 32, 64 × 64, 96 × 96, and 128 × 128), using PyRadiomics, version 3.0.1. A total of 93 features, including first-order features and texture features, were extracted from each patch. In total, 372 features (93 for each scale) were extracted for the image after conversion via hue match. The constant features were removed in the first step of feature selection. In the second step, the feature pairs with Spearman’s correlation coefficient (|r| > 0.90) were deemed as highly correlated, and the feature with the highest average correlation with all other features was removed. Recursive feature elimination (RFE) based on a random forest (RF) classifier was used as a final step for feature selection. A random forest classifier-based radiomics model was trained using the selected radiomics features to predict the probability of symptomatic PHLF in terms of the radiomics score. A five-fold cross-validation was used to fine-tune the hyperparameters. For the patient-level analysis, the median radiomics score of all the images from one patient was considered to be the radiomics score for that patient.


(3)The clinical–radiomics model based on 2D-SWE images and clinical data:


Univariate and multivariate logistic analyses were performed in the training cohort to identify independent clinical predictors of symptomatic PHLF. A logistic regression model clinical–radiomics based on the radiomics score and significant clinical variables was constructed for symptomatic PHLF prediction.

### 2.6. Shapley Additive exPlanations

SHAP is a post hoc interpretability method that is based on game theory, and it was used for understanding the predictions made by the radiomics model. It measures the importance of each feature and its effect on the model’s predicted probability in terms of SHAP values [22]. SHAP summary plots provide global explanations by quantifying the impact of feature values on the model output and help in identifying the important features and their trends. SHAP dependence plots show how the model is affected by an individual feature. These dependence plots also show interaction effects between a pair of features and their resulting impact on the model output. SHAP local bar plots display the SHAP values for an individual test example, showing the impact of each feature on the model outcome.

### 2.7. Statistical Analysis

Statistical analyses were performed by SPSS, version 20.0. Student’s *t*-test or the Mann–Whitney test, as appropriate, was used to compare the continuous variables in the training and test cohorts. The χ2 test was used to compare categorical variables. A two-sided *p* < 0.05 means that the corresponding estimate reaches a significant difference. A univariate logistic analysis was performed in the training cohort to detect significant predictors associated with symptomatic PHLF. These variables entered a stepwise multivariate logistic regression analysis to identify independent predictors for symptomatic PHLF. The clinical model was established based on independent predictors by logistic regression. Open-source Python v3.6.13 was used to implement the radiomics analysis. A detailed description of the packages and versions is given in Appendix A. The AUCs were compared using the DeLong test between different models. The thresholds of each model were set at the highest Youden index in the training cohort. The patient-level performance metrics, including the accuracy, sensitivity, specificity, positive predictive value (PPV), and negative predictive value (NPV), of the models were evaluated and reported. A nomogram was constructed based on the clinical–radiomics model. Calibration curves were plotted to analyze the calibration performance of the different models in the test set. A decision curve analysis was conducted in the test set to determine the clinical usefulness of the nomogram by quantifying the net benefits at different threshold probabilities.

## 3. Results

### 3.1. Baseline Characteristics

A total of 345 patients were enrolled, of which 305 were males and 40 were females, with a median age of 55.0 (IQR 47.0–64.0) years (Figure 1). There were 265 patients in the training cohort and 80 patients in the test cohort.

The baseline characteristics of the training and test cohorts were summarized in Table 1. A total of 107 patients (31.0%) experienced symptomatic PHLF, including 97 patients with PHLF grade B and 10 patients with PHLF grade C. Six patients with PHLF grade C died of acute liver failure within 20 to 39 days after surgery. Symptomatic PHLF was observed in 80 (30.1%) patients and 27 (33.8%) patients in the training and test cohorts, respectively, showing no significant difference. There were significant differences in the prothrombin time (PT) level (*p* = 0.002), international normalized ratio (INR) level (*p* < 0.001), and MELD score (*p* = 0.012) between the training and test cohorts.

### 3.2. Performance of the Clinical Model

The multivariate logistic regression analysis showed that the INR, CSPH, and LRV ratio were significant independent predictors of symptomatic PHLF (all *p* < 0.05; Table 2). These three variables were included to establish the clinical model. The clinical model showed an AUC of 0.809 (95% CI: 0.715–0.902) and 0.684 in the five-fold cross-validation and the test cohort, respectively.

### 3.3. Performance of the Radiomics Model and the Clinical–Radiomics Model in Five-Fold Cross-Validation

In the five-fold cross-validation, the radiomics model with hue match showed a higher AUC (0.741; 95% CI: 0.662–0.819) than models with other RGB-to-gray conversion methods, including distance match, RGB three channels, and direct conversion (AUC: 0.728–0.738) (Figure 4a). The radiomics model combining seven radiomics features from different patches showed a better performance (AUC: 0.746; 95% CI: 0.681–0.811) than models with a single patch of ROI (AUC: 0.726–0.741) (Figure 4b). So, the multi-patch radiomics model with hue match was adopted in this study to develop a radiomics model. The clinical–radiomics model which combined the radiomics score and clinical features achieved an AUC of 0.867 (95% CI: 0.787–0.947), which was higher than the clinical model (AUC: 0.809; 95% CI: 0.715–0.902) and radiomics model (AUC: 0.746; 95% CI: 0.681–0.811) (Figure 5a and Table 3).

### 3.4. Performance of the Radiomics Model and the Clinical–Radiomics Model in the Test Set

In the test set, the AUC, accuracy, sensitivity, specificity, PPV, and NPV of the radiomics model were 0.784 (95% CI: 0.720–0.898), 0.725, 0.660, 0.754, 0.581, and 0.816, respectively (Table 4). The AUC, accuracy, sensitivity, specificity, PPV, and NPV of the clinical–radiomics model were 0.822 (95% CI: 0.720–0.898), 0.750, 0.704, 0.773, 0.612, and 0.836, respectively (Table 4). The clinical–radiomics model showed a significantly higher AUC than the clinical model (AUC: 0.684, *p* = 0.007), as well as some clinical indices related to symptomatic PHLF prediction, such as the MELD score (AUC: 0.529, *p* < 0.001) and ALBI score (AUC: 0.644, *p* = 0. 016) (Figure 5b). The clinical–radiomics model showed a higher AUC than the radiomics model (AUC: 0.784, *p* = 0.415), without significant difference. The nomogram of the clinical–radiomics model is shown in Figure 6a. Good calibration was achieved for the clinical–radiomics model in the test set (Figure 6b), and the decision curve for the clinical–radiomics model showed a higher net benefit for the clinical–radiomics model than for the clinical model and the radiomics model when the threshold probability was between 0.10 and 0.58 (Figure 6c).

### 3.5. Shapley Additive exPlanations

The global SHAP summary plot identified the first-order maximum 64 × 64, first-order 90th percentile 64 × 64, and first-order 10th percentile 32 × 32 as the most important features for symptomatic PHLF prediction. These features had a similar trend: a higher feature value resulted in a high positive SHAP value (Figure 7a), which corresponded with higher predicted probability. The fourth important feature was the gray-level co-occurrence matrix Informational Measure of Correlation (glcm_Imc1 96 × 96), and it had a negative trend: a higher feature value resulted in a lower negative SHAP value. SHAP dependence plots of first-order maximum 64 × 64 and first-order 90th percentile 64 × 64 showed the relationship between the SHAP values and the feature values, as well as the interaction with another feature (Figure 7b, c). A higher value for the first-order 90th percentile 64 × 64 resulted in a higher SHAP value. However, when the glcm_Imc1 96 × 96 value was also high, the SHAP value was comparatively lower (Figure 7c). Figure 7d, e show the SHAP local bar plots for two test cases that had symptomatic PHLF. Figure 7d shows a case that was classified correctly, and the plot shows that all features except glcm_Imc1 96 × 96 made the correct contribution. Figure 7e shows a case that has been classified incorrectly by the model, and the plots show that only first-order maximum 64 × 64 and first-order minimum 32 × 32 made the correct contribution.

## 4. Discussion

In this study, we proposed an interpretable clinical–radiomics model based on liver 2D-SWE images and clinical variables for the prediction of symptomatic PHLF in HCC patients. The clinical–radiomics model achieved an AUC of 0.822 in the test cohort, which was higher than that of the clinical model and some clinical variables, including the ALBI score and MELD score. A nomogram was established of the clinical–radiomics model for clinical use. The SHAP analysis showed that first-order statistical features were most important for model prediction, which confirmed the reliability of the developed radiomics model and helped clinicians understand the model. The results showed that a radiomics analysis of 2D-SWE images may serve as a useful tool to stratify high-risk and low-risk patients for symptomatic PHLF and to assist the surgeons in recognizing the best candidates for liver resection, determining the resection extent, and improving perioperative management.

Several studies have verified the utility of 2D-SWE for predicting symptomatic PHLF, with AUCs ranging from 0.72 to 0.76 [13,30]. In this study, the radiomics method enabled a comprehensive analysis of 2D-SWE images and showed a better performance. The performance of the clinical–radiomics model developed in our study was higher than that of other reported predictive models, with AUCs ranging from 0.72 to 0.82 [31,32,33].

For the radiomics analysis, we proposed a multi-patch strategy that extracted coarse-to-fine radiomics features and resulted in better predictive accuracy than the single-patch strategy. This result was consistent with another study showing that a multi-patch texture features analysis of ultrasound images led to better performance for liver fibrosis grading than the single-patch analysis [34]. In our study, five first-order features from patches of 32 × 32 pixels and 64 × 64 pixels and two texture features from patches of 96 × 96 pixels were selected. The first-order features from smaller patches might be more informative since they avoided artifacts and noise areas within 2D-SWE images; this result was consistent with another study showing that the automatic selection of the most homogenous area for ROI improved the accuracy of liver fibrosis staging [29]. Texture features from larger patches may be more informative because they are sensitive to global texture features. The clinical–radiomics model outperformed both the clinical model and radiomics model, suggesting that radiomics features and clinical features were complementary to each other.

Furthermore, a SHAP analysis was performed to understand the contribution of each radiomics feature to the radiomics signature. The global SHAP analysis identified first-order features as the most important features for symptomatic PHLF prediction, which was quite explainable because higher first-order statistical features corresponded with higher liver stiffness, therefore leading to a higher probability of symptomatic PHLF. The results were consistent with existing studies showing that higher liver stiffness was correlated with symptomatic PHLF [12,13,30], which confirmed the reliability of the developed radiomics score.

The strengths of the radiomics analysis applied in this study were as follows. Firstly, we applied a new multi-patch strategy for radiomics analysis, which could be an efficient method for a radiomics analysis of 2D-SWE images in future studies. Secondly, we effectively combined the high-throughput 2D-SWE features with low-dimensional clinical information, which demonstrated a better predictive performance for symptomatic PHLF prediction. Thirdly, the SHAP analysis was used to improve the interpretability of the complex classifiers in radiomics, helping clinicians understand the models. Fourthly, compared with LSM, the developed multi-patch radiomics strategy fully leverages all the information contained within 2D-SWE images. Moreover, it effectively mitigates inter-observer variances, offering a more automatic, objective, and comprehensive approach.

This study has some limitations. The significant differences in the PT level, INR level, and MELD score between the training and test cohorts could potentially affect the predictive performance in the test cohort. It is a single-center study, so multicentric external validation is needed to verify its generalizability before taking steps towards clinical application. In addition, 94% of the patients enrolled were infected with hepatitis B. Therefore, the performance of the radiomics model on patients with other causes of underlying liver diseases needs further study.

## 5. Conclusions

In conclusion, the clinical–radiomics model based on 2D-SWE images and clinical variables was useful for predicting symptomatic PHLF in HCC with high model interpretability. It may serve as a useful tool for therapeutic decision making to improve perioperative management. Further prospective multicenter studies and patients with different etiologies should be considered to validate and optimize the model.

## Figures and Tables

**Figure 1 cancers-15-05303-f001:**
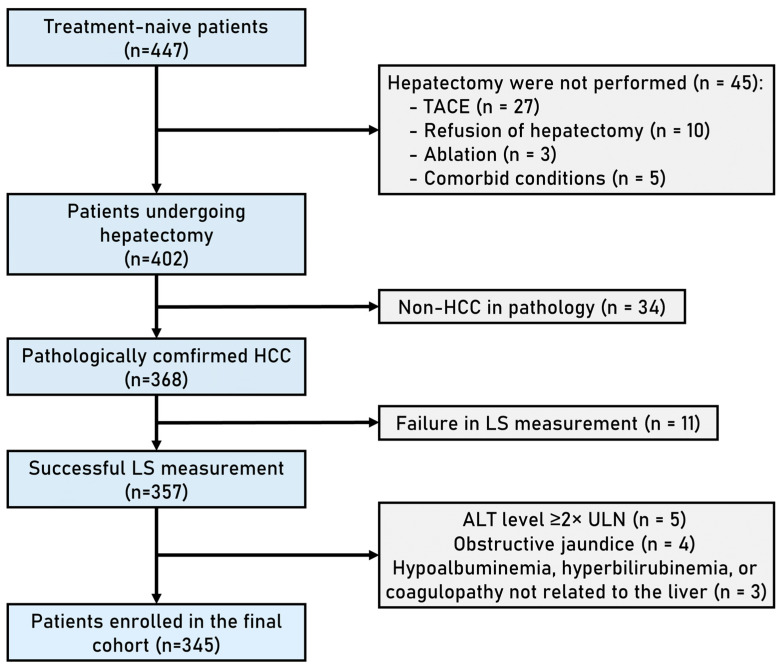
Flowchart of patient enrollment.

**Figure 2 cancers-15-05303-f002:**
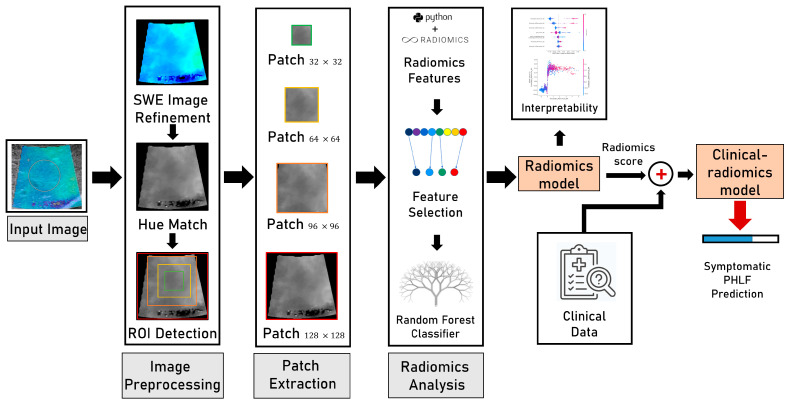
The workflow of constructing radiomics models.

**Figure 3 cancers-15-05303-f003:**
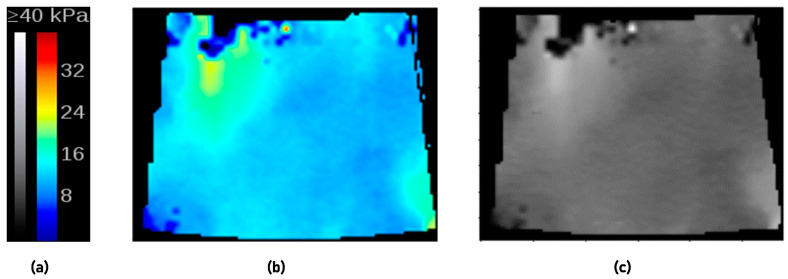
(**a**) The color bar. (**b**) Color elasticity image before the transformation. (**c**) Grayscale image after the hue match transformation of color elasticity image.

**Figure 4 cancers-15-05303-f004:**
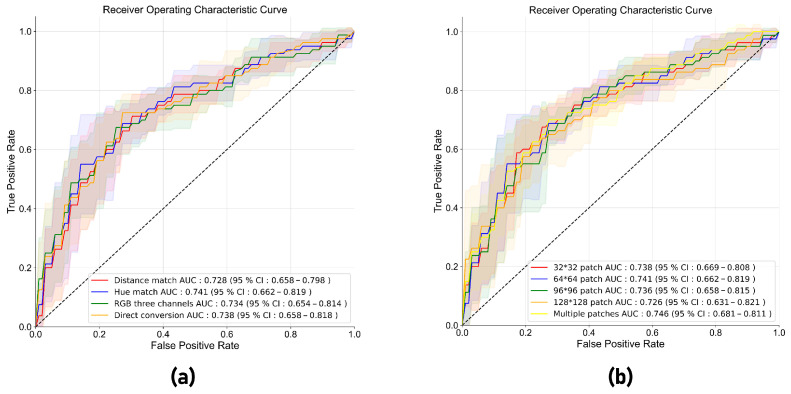
(**a**) Receiver operating characteristic curves of radiomics models of four different image preprocessing methods (distance match, hue match, RGB three channels, and direct conversion) based on the 64 × 64 pixel patch in five-fold cross-validation. (**b**) ROC curves of radiomics models of different patches (32 × 32, 64 × 64, 96 × 96, and 128 × 128 pixels, as well as the combination of multiple patches) based on hue-match preprocessing in five-fold cross-validation.

**Figure 5 cancers-15-05303-f005:**
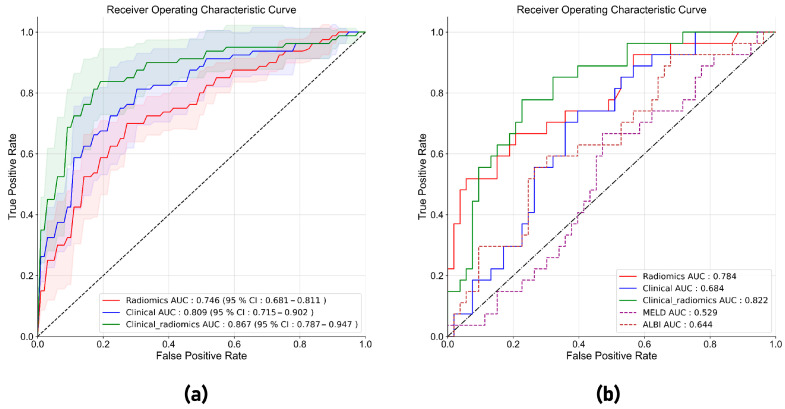
(**a**) Receiver operating characteristic curves for radiomics model, clinical model and clinical–radiomics model in five-fold cross-validation. (**b**) Receiver operating characteristic curves for radiomics model, clinical model, clinical–radiomics model and other clinical indices the test cohort.

**Figure 6 cancers-15-05303-f006:**
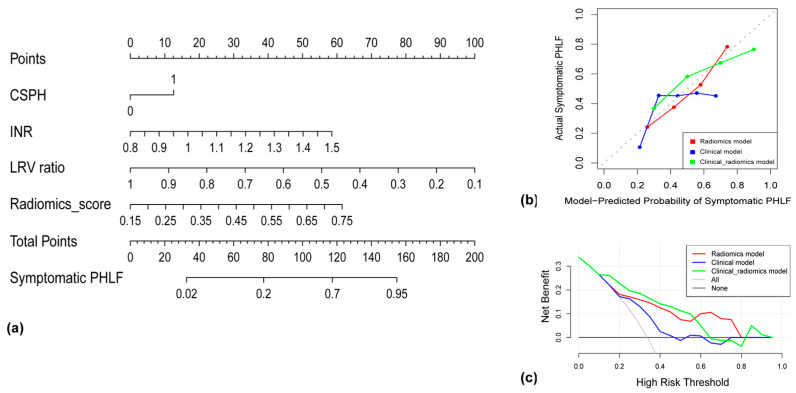
(**a**) Nomogram for prediction of symptomatic PHLF. CSPH, clinically significant portal hypertension; INR, international normalized ratio; LRV ratio, ratio of future liver remnant volume; PHLF, post-hepatectomy liver failure. (**b**) Calibration curves of the radiomics model, the clinical model, and the clinical–radiomics model in the test set. (**c**) Decision curve analysis of the radiomics model, the clinical model, and the clinical–radiomics model in the test set.

**Figure 7 cancers-15-05303-f007:**
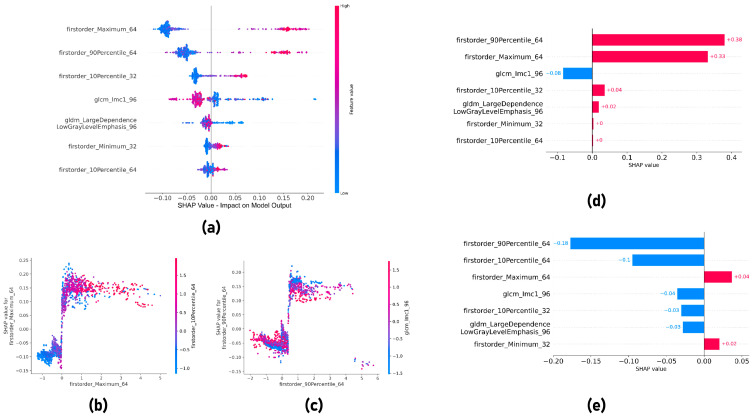
(**a**) SHAP global summary plot. SHAP dependence plots for (**b**) first-order maximum 64 × 64 and (**c**) first-order 90th Percentile 64 × 64. (**d**,**e**) SHAP local bar plots for two test cases that have symptomatic PHLF.

**Table 1 cancers-15-05303-t001:** Baseline characteristics of enrolled patients.

Characteristic	All Patients (*n* = 345)	Training Cohort (*n* = 265)	Test Cohort (*n* = 80)	*p*-Value
Age (year)	55.0 (47.0–64.0)	55.0 (47.0–64.0)	54.0 (49.0–66.8)	0.354
Sex (male/female)	305/40	238/27	67/13	0.138
Underlying liver disease (HBV/HCV/coinfection of HBV and HCV/unknown)	324/7/6/8	249/5/5/6	75/2/1/2	0.965
TBIL (umol/L)	13.8 (10.7–17.3)	13.6 (10.6–16.9)	15.0 (11.5–17.9)	0.081
ALB (g/L)	38.3 (36.2–41.0)	38.3 (36.2–41.0)	38.8 (36.2–41.2)	0.942
CREA (umol/L)	79.0 (68.0–87.0)	79.0 (68.0–87.5)	80.0 (68.3–87.0)	0.940
ALT (U/L)	31.0 (21.0–43.5)	32.0 (20.0–43.0)	31.0 (22.0–52.8)	0.805
AST (U/L)	35.0 (25.0–50.0)	35.0 (26.0–50.0)	36.0 (23.0–50.7)	0.812
GGT (U/L)	55.0 (34.0–98.5)	59.0 (36.0–103.0)	50.0 (30.3–85.8)	0.055
PT (s)	11.9 (11.3–12.6)	11.8 (11.2–12.4)	12.2 (11.7–12.8)	0.002
INR	1.02 (0.97–1.07)	1.01 (0.96–1.06)	1.05 (1.00–1.09)	<0.001
AFP (U/L)	23.1 (4.4–516.1)	21.2 (4.5–527.4)	49.6 (4.1–476.3)	0.829
ALBI	−2.52 [(−2.72)–(−2.34)]	−2.53 [−(2.73)–(−2.34)]	−2.54 [−(2.67)–(−2.33)]	0.863
ALBI grade (1/2)	137/208	104/161	33/47	0.748
Child–Pugh score (5/6/7)	276/53/16	211/39/15	65/14/1	0.234
Child–Pugh grade (A/B)	329/16	250/15	79/1	0.100
MELD	4.8 (2.9–6.3)	4.6 (2.6–6.2)	5.4 (3.9–7.3)	0.012
Cirrhosis (yes/no)	120/225	90/175	30/50	0.560
CSPH (yes/no)	39/306	29/236	10/70	0.700
Splenomegaly (yes/no)	101/244	83/182	18/62	0.129
Ascite (yes/no)	22/323	19/246	3/77	0.273
Tumor size (cm)	5.4 (3.5–8.3)	5.7 (3.6–8.4)	4.5 (3.0–7.5)	0.107
BCLC stage (0/A/B/C)	19/222/62/42	14/163/49/39	5/59/13/3	0.051
TLV (mL)	1242.4 (1083.4–1528.2)	1242.4 (1086.1–1531.2)	1230.6 (1070.5–1526.0)	0.707
RLV (mL)	428.0 (234.7–687.5)	433.1 (234.7–704.9)	366.1 (226.1–633.3)	0.229
LRV	788.6 (643.5–963.1)	788.6 (631.6–957.7)	787.2 (689.9–1003.6)	0.465
LRV ratio	0.67 (0.50–0.80)	0.66 (0.48–0.79)	0.69 (0.54–0.80)	0.186
Symptomatic PHLF (yes/no)	107/238	80/185	27/53	0.546

Continuous variables are expressed in median (P25–P75). Categorical variables are expressed in counts. TBIL, total bilirubin; ALB, albumin; CREA, creatinine; ALT, alanine aminotransferase; AST, aspartate transaminase; GGT, gamma-glutamyl transferase; PT, prothrombin time; INR, international normalized ratio; AFP, alpha-fetoprotein; ALBI, albumin–bilirubin; MELD, model for end-stage liver disease; CSPH, clinically significant portal hypertension; BCLC, Barcelona Clinic Liver Cancer; TLV, total liver volume; RLV, resected liver volume; LRV, future liver remnant volume; PHLF, post-hepatectomy liver failure.

**Table 2 cancers-15-05303-t002:** Influencing clinical factors of symptomatic PHLF.

Variables	Univariate Analysis	*p*-Value	Multivariate Analysis	*p*-Value
	OR (95% CI)		OR (95% CI)	
Sex, female vs. male	0.791 (0.320–1.953)	0.791	−	−
Age (years)	1.003 (0.981–1.025)	0.811	−	−
TBIL (umol/L)	1.033 (0.998–1.069)	0.062	−	−
ALB (g/L)	0.900 (0.838–0.967)	0.004	−	−
CREA (umol/L)	0.997 (0.988–1.006)	0.562	−	−
ALT (U/L)	1.001 (0.998–1.005)	0.446	−	−
AST (U/L)	1.002 (0.999–1.006)	0.234	−	−
GGT (U/L)	1.003 (1.000–1.005)	0.024	−	−
PT (s)	1.343 (1.058–1.706)	0.015	−	−
INR	2461.350 (70.906–85,440.280)	<0.001	2424.484 (49.342–119,130.427)	<0.001
AFP (U/L)	1.000(1.000–1.000)	0.244	−	−
ALBI score	4.533 (1.947–10.557)	<0.001	−	−
Child–Pugh score	2.031 (1.292–3.193)	0.002	−	−
Child–Pugh grade, B vs. A	3.782 (1.299–11.013)	0.015	−	−
MELD	1.115(1.018–1.222)	0.019	−	−
Cirrhosis, yes vs. no	2.499 (1.450–4.307)	0.001	−	−
CSPH, yes vs. no	3.308 (1.507–7.260)	0.003	4.670 (0.001–0.023)	0.001
Splenomegaly, yes vs. no	1.618 (0.931–2.811)	0.088	−	−
Ascites, yes vs. no	2.218 (0.865–5.689)	0.097	−	−
Tumor size (cm)	1.177 (1.090–1.272)	<0.001	−	−
BCLC stage	1.536 (1.113–2.118)	0.009	−	−
TLV (mL)	1.001 (1.000–1.002)	0.001	−	−
RLV (mL)	1.002 (1.001–1.003)	<0.001	−	−
LRV (mL)	0.997 (0.996–0.998)	<0.001	−	−
LRV ratio	0.009 (0.002–0.039)	<0.001	0.004 (0.001–0.023)	<0.001

TBIL, total bilirubin; ALB, albumin; CREA, creatinine; ALT, alanine aminotransferase; AST, aspartate transaminase; GGT, gamma-glutamyl transferase; PT, prothrombin time; INR, international normalized ratio; AFP, alpha-fetoprotein; ALBI, albumin–bilirubin; MELD, model for end-stage liver disease; CSPH, clinically significant portal hypertension; BCLC, Barcelona Clinic Liver Cancer; TLV, total liver volume; RLV, resected liver volume; LRV, future liver remnant volume.

**Table 3 cancers-15-05303-t003:** Five-fold cross-validation results.

Model	AUC (CI)	Accuracy ±STD	Sensitivity ±STD	Specificity ±STD	PPV ±STD	NPV ±STD
Radiomics	0.746 (0.681–0.811)	0.698 ± 0.054	0.725 ± 0.064	0.686 ± 0.098	0.511 ± 0.060	0.853 ± 0.019
Clinical	0.809 (0.715–0.902)	0.739 ± 0.051	0.713 ± 0.170	0.751 ± 0.035	0.549 ± 0.057	0.865 ± 0.078
Clinical–radiomics	0.867 (0.787–0.947)	0.800 ± 0.081	0.800 ± 0.0.073	0.800 ± 0.103	0.652 ± 0.120	0.901 ± 0.035

AUC, area under the receiver operating characteristic curve; PPV, positive predictive value; NPV, negative predictive value.

**Table 4 cancers-15-05303-t004:** Test set results.

Model	AUC	Accuracy	Sensitivity	Specificity	PPV	NPV
Radiomics	0.784	0.725	0.660	0.754	0.581	0.816
Clinical	0.684	0.650	0.550	0.698	0.484	0.755
Clinical–radiomics	0.822	0.750	0.704	0.773	0.612	0.836

AUC, area under the receiver operating characteristic curve; PPV, positive predictive value; NPV, negative predictive value.

## Data Availability

The datasets used during the current study are available from the corresponding author upon reasonable request.

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
