# Peer review of "An Interpretable Radiomics Model Based on Two-Dimensional Shear Wave Elastography for Predicting Symptomatic Post-Hepatectomy Liver Failure in Patients with Hepatocellular Carcinoma"

_cancers, 2023, doi:10.3390/cancers15215303_

Round 1

Reviewer 1 Report

Comments and Suggestions for Authors

Dear authors,

I read with great interest the manuscript entitled “An Interpretable Radiomics Model Based on Two-dimensional Shear Wave Elastography for Predicting Symptomatic Post- 3 hepatectomy Liver Failure in Patients with Hepatocellular Carcinoma ”.

The authors conducted a study on 345 consecutive patients on a radiomics model to predict for predicting symptomatic post-hepatectomy liver failure in HCC patients.

The paper is well constructed and presented.

I have no futher comments.

Author Response

We feel great thanks for your valuable comments concerning our manuscript.

Reviewer 2 Report

Comments and Suggestions for Authors

The authors developed and validated an interpretable radiomics model based on two-dimensional shear wave elastography (2D-SWE) for predicting symptomatic post-hepatectomy liver failure (PHLF) in patients undergoing liver resection for hepatocellular carcinoma. There is strong guiding significance for clinical practice. The study shows promising, but I believe addressing the mentioned issues will significantly enhance the quality and clarity of the manuscript. Followed are my comments: 

1. Previous studies have also shown the potential value of liver stiffness measurement (LSM) with 2D-SWE in predicting PHLF, what is the point of innovation in your radiomics model?please provide more information about what is your strength in your study?

2. As for the location of liver area, why the right anterior lobe of the liver was preferred?

3. There were significant differences in prothrombin time (PT) level, international normalized ratio (INR) level and MELD score between the training and test cohorts, all of which were risk clinical factors for PHLF. Did those factors affect the predictive performance in test cohort? 

Comments on the Quality of English Language

 Minor editing of English language required.

Reviewer 3 Report

Comments and Suggestions for Authors

In this study, Dr. Zhong and his co-workers aimed to develop and validate an interpretable radiomics model based on two-dimensional shear wave elastography (2D-SWE) for predicting symptomatic post-hepatectomy liver failure (PHLF) in patients undergoing liver resection for hepatocellular carcinoma (HCC). They demonstrated that the clinical-radiomics model performed better than radiomics model and clinical model, and some clinical variables including ALBI score and MELD score.

I congratulate their excellent work. It was a very helpful study, and well written in English. A few minor problems in the manuscript should be noticed before considering publication.

1. In lines 219-221, the authors employed a random forest classifier to construct a radiomics model. I would like to understand the rationale behind choosing this particular method when there are numerous other techniques available. How does it compare to Convolutional Neural Networks, a powerful tool for image processing? Could you please elucidate the basis for this choice?

2. In lines 275-277, the authors mentioned that ‘There were significant differences in prothrombin time (PT) level (p = 0.002), international normalized ratio (INR) level (p <  0.001) and MELD score (p = 0.012) between the training and test cohorts.’ Given that INR is one of the significant independent predictors of symptomatic PHLF, and these baseline clinical features appear to be unbalanced, could this unbalanced distribution potentially impact the reliability of the results?"

3. In lines 380-381, the authors concluded that 'The clinical-radiomics model achieved an AUC of 0.822 in the test cohort, which was higher than the radiomics model and clinical model, and some clinical variables including ALBI score and MELD score.' However, it's important to note that while the AUC of the clinical-radiomics model was higher than that of the radiomics model in the test set, no statistically significant difference was observed between the two (P = 0.451). Therefore, the statement may need to be reconsidered as it may lack a strong statistical basis.
